# Cross-Lingual Consistency of Factual Knowledge in Multilingual Language Models

**Jirui Qi** [1], **Raquel Fernández**[2], **Arianna Bisazza**[1]

[1] Center for Language and Cognition, University of Groningen
[2] Institute for Logic, Language and Computation, University of Amsterdam
{j.qi, a.bisazza}@rug.nl raquel.fernandez@uva.nl

## Abstract

Multilingual large-scale Pretrained Language Models (PLMs) have been shown to store considerable amounts of factual knowledge, but large variations are observed across languages. With the ultimate goal of ensuring that users with different language backgrounds obtain consistent feedback from the same model, we study the cross-lingual consistency (CLC) of factual knowledge in various multilingual PLMs. To this end, we propose a Ranking-based Consistency (RankC) metric to evaluate knowledge consistency across languages independently from accuracy. Using this metric, we conduct an in-depth analysis of the determining factors for CLC, both at model level and at language-pair level. Among other results, we find that increasing model size leads to higher factual probing accuracy in most languages, but does not improve cross-lingual consistency. Finally, we conduct a case study on CLC when new factual associations are inserted in the PLMs via model editing. Results on a small sample of facts inserted in English reveal a clear pattern whereby the new piece of knowledge transfers only to languages with which English has a high RankC score.[1]

## 1 Introduction

Large-scale Pre-trained Language Models (PLMs) have demonstrated powerful capabilities in tasks where factual knowledge plays an important role (Roberts et al., 2020; Qin et al., 2022). While most previous work on probing factual knowledge in PLMs has focused on English (Davison et al., 2019; Bouraoui et al., 2020; Shin et al., 2020; Brown et al., 2020; Alghanmi et al., 2021; Peng et al., 2022), a few notable studies have extended the evaluation to a number of other languages (Jiang et al., 2020; Kassner et al., 2021; Yin et al., 2022). The results of these studies show a large variation in

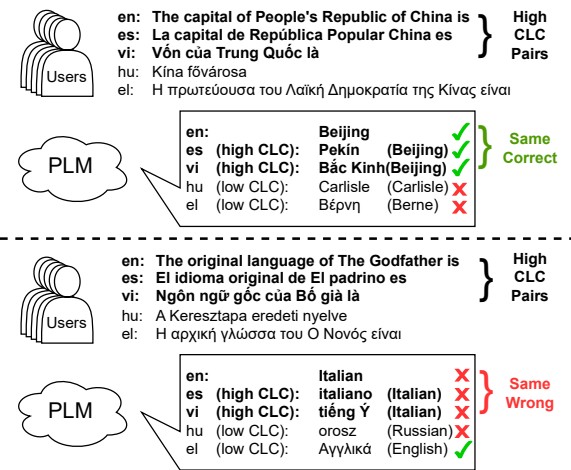

Figure 1: Motivating example: Some languages share consistent knowledge in the multilingual PLM BLOOM-3b, while others do not.

the extent to which factual knowledge generalizes across languages, revealing yet another facet of language inequality in modern NLP technologies (Hupkes et al., 2022).

Assessing factual knowledge across languages, however, is not a trivial endeavor. Ensuring comparability of the results requires that a single set of 'universal' facts is queried in all languages, but the choice of that set is likely to be biased to specific world regions that are more represented in popular knowledge bases like Wikidata.[2] Conversely, facts that are more relevant in other regions of the world (e.g., information about locations or important persons in a given region) are less likely to occur in the benchmarks, which makes it hard to interpret the results of such evaluations.

In this work, we take a different stance: instead of measuring the *amount* of factual knowledge encoded by a PLM in each language, we focus on its *consistency* across languages. As illustrated in Figure 1, the multilingual BLOOM-3b model (Scao

---

[1] All code and data released at https://github.com/Betswish/Cross-Lingual-Consistency

[2] https://www.wikidata.org

et al., 2022) outputs consistently correct completions of the first prompt when queried in English, Spanish, and Vietnamese, but not in Hungarian and Greek. The model also outputs consistent, though wrong, answers to the second query in English, Spanish, and Vietnamese (but not in Hungarian and Greek), suggesting the first three languages share relevant knowledge representations within the model.

The study of cross-lingual consistency (CLC) is important for at least two reasons: Firstly, true knowledge of a fact implies encoding of its meaning regardless of a given surface form (Ohmer et al., 2023). Thus, if a model knows that the city of Beijing is the capital of China, it should return the same answer when asked the same question in different languages. From a practical point of view, CLC is crucial to ensure users have a similar experience when interacting with the same model in different languages. Secondly, studying CLC is important to understand whether and how knowledge acquired in a language gets implicitly transferred to another within multilingual PLMs. Besides scientific relevance, this has practical implications for the incorporation of external knowledge into multilingual PLMs. In fact, while a prolific line of work has focused on *model editing* as a way to insert new factual associations in PLMs in various data- and computation-efficient manners (De Cao et al., 2021; Hou et al., 2022; Meng et al., 2022), no one, to our knowledge, has looked yet at how this affects the factual knowledge in languages other than the one to which editing is directly applied.

We conduct the first in-depth study of CLC of factual knowledge in multilingual PLMs and make the following contributions: (i) We propose a novel Ranking-based Consistency (RankC) metric which assesses knowledge consistency independently from accuracy. (ii) We filter the existing unbalanced datasets (Jiang et al., 2020; Kassner et al., 2021) to form a multi-parallel CLC benchmark, Balanced Multilingual LAnguage Model Analysis (BMLAMA), which has the same set of prompts translated into all languages. (iii) We apply the new metric to BMLAMA to assess CLC in various encoder-only, decoder-only, and encoder-decoder PLMs, including XLM-RoBERTa-large, mT5-large, and BLOOM series. We analyze a number of language properties that correlate with CLC and provide new insights into how factual knowledge percolates among languages. Finally (iv) we

present a case study with a state-of-the-art model editing technique based on neuron interpretability (Meng et al., 2022), providing preliminary evidence that CLC is predictive of whether a fact inserted in language X will transfer to language Y.

## 2 Related Work

**Probing factual knowledge in PLMs** Since first proposed by LAMA (Petroni et al., 2019), prompt-based probing has become the main technique to assess factual knowledge in PLMs (Davison et al., 2019; Bouraoui et al., 2020; Shin et al., 2020; Brown et al., 2020; Alghanmi et al., 2021; Peng et al., 2022). Given the knowledge represented in a tuple (*subject*, *relation*, *object*), a query $q$ is formed by filling the *subject* into a *relation*-specific template, which is fed into the PLM. If the prediction is in line with the *object*, the model is considered to possess this knowledge. For instance, given a candidate set of city names, when queried with 'The *capital* of *People's Republic of China* is _', the PLM is considered to capture this piece of knowledge if it gives the highest probability to the correct answer '*Beijing*', among all candidates.

**Multilingual probing of factual knowledge** Next to a multitude of works focusing on English, a few notable studies have probed factual knowledge multilingually by translating the English prompt-object pairs into a number of languages. X-FACTR (Jiang et al., 2020) and MLAMA (Kassner et al., 2021) indicate strong variations between the amount of knowledge in different languages, due to the size of their training corpora. Apart from English and a handful of other high-resource European languages, very low (i.e. <10%) probing accuracies are reported overall. Another relevant work, GeoMLAMA (Yin et al., 2022), probes specifically commonsense knowledge that is susceptible to vary across different regions, leading to the rather surprising finding that the best language to probe knowledge about a certain country (e.g. China) is often not the native language of the given country (e.g. Chinese). The main focus of all these studies is on assessing the amount of factual knowledge encoded for each language, rather than on understanding how such knowledge percolates among languages.

**Self-consistency** Self-consistency refers to a PLM's ability to output the same answer to meaning-preserved paraphrases of the same query.

Self-consistency of English PLMs has received attention across different tasks (Li et al., 2019; Mitchell et al., 2022; Wang et al., 2023). Fierro and Søgaard (2022) extend the study of self-consistency to multilingual PLMs, by measuring it separately in each language. Their results show poor self-consistency in all languages.

**Cross-lingual consistency** To our knowledge, we are the first to conduct a systematic analysis of *cross-lingual* consistency of factual knowledge in multilingual PLMs, that is the extent to which a PLM returns the same answer to the same question asked in different languages. As part of their probing study, Jiang et al. (2020) compute the ratio of *correct* predictions that overlap between two languages within mBERT (cf. Sect. 3.1). They report low ratios overall, with a peak of only 34% in the most similar pair (English-Dutch), but do not further investigate the factors that determine consistency. Moreover, they limit this analysis to one (encoder-only) model while we also examine an encoder-decoder and a series of decoder-only models (cf. Sect. 5.1). As another difference, we take a more holistic view of consistency, whereby predictions that are incorrect but refer to the same entity across languages should also be considered consistent. Interestingly, concurrent work by Ohmer et al. (2023) proposes to use the cross-lingual consistency of a model's predictions as a means to assess its understanding of meaning beyond specific word forms. They showcase their approach in two language understanding tasks (paraphrase identification and natural language inference). Despite the different scopes, their evaluation of ChatGPT using English, German, and Chinese translations reveals limited consistency in the model responses, which is in line with our factual probing findings (cf. Sect. 5) and further indicates this problem persists in very large-scale, last-generation PLMs.

## 3 Measuring Cross-Lingual Consistency

**Task definition** Each language $l \in \mathcal{L}$ has a set of queries (i.e. prompts) defined as $Q_l$. For each query $q_i \in Q_l$, there are $N_i$ corresponding candidates (Kassner et al., 2021; Yin et al., 2022). For example, the query '*Steve Jobs worked for __*' has 10 candidates: *Apple, Nintendo, Google, WWE, Alexandria, Germany, Yahoo, Berlin, BBC, Microsoft*. Each query is fed to the PLM, and the returned probabilities are used to compute a ranking score for each of the candidate words. The specific

score calculation depends on the type of model (encoder-only, encoder-decoder, or decoder-only) and on the way a candidate word gets segmented into subwords (see details in Appendix B). After being sorted by ranking scores, the candidate set for $q_i$ is represented as $\{c_i^1, \ldots, c_i^{N_i}\}$, where $c_i^1$ has the highest prediction probability and $c_i^{N_i}$ has the lowest. Note that the existing multilingual datasets for knowledge probing (X-FACTR (Jiang et al., 2020) and MLAMA (Kassner et al., 2021)) have different numbers of queries in different languages, which is problematic for measuring consistency.

### 3.1 Prior Work: Correct Predictions Overlap

Based on the predictions $c_i^1$ and $c'^1_i$ (i.e. first elements of the sorted candidate lists) of each $q_i$ and $q'_i$, Jiang et al. (2020) compute the average overlapping ratio of correct predictions as follows:

$$\text{COverlap}(l, l') = \frac{\sum_{i=1}^{|Q_l^*|} \mathbb{1}(c_i^1 = o_i \& c'^1_i = o'_i)}{\sum_{i=1}^{|Q_l^*|} \mathbb{1}(c_i^1 = o_i \| c'^1_i = o'_i)} \quad (1)$$

where $\mathbb{1}(\cdot)$ is an indicator function, and $o_i$ and $o'_i$ are the correct answer for $q_i$ and $q'_i$ respectively.

Because their benchmark contains different amounts of queries in different languages, they filter the query set for each language pair $(l, l')$ by discarding samples that are not available in either $l$ or $l'$:

$$Q_l^*, Q_{l'}^* = filter(Q_l, Q_{l'})$$
$$|Q_l^*| = |Q_{l'}^*| \quad (2)$$

Since the filtering is done on each language pair separately, this results in different query sets, which limits the comparability of their results across language pairs that have very different filtered sets.

### 3.2 This Work: RankC Metric

To ensure comparability between different language pairs, we instead require that *all* queries in our benchmark and their corresponding candidates are translated across all languages. Thus, for any language pair $(l, l')$, the length of the query sets is always equal $|Q_l| = |Q_{l'}|$, and so is the number of candidates for the $i$-th query $N_i = N'_i$.

Based on these assumptions, we propose a novel Ranking-based Consistency (RankC) metric to effectively assess the cross-lingual consistency of knowledge in PLMs, independently from accuracy. Instead of merely focusing on the correct predictions, we take the rankings of all candidates into

consideration. RankC is inspired by the Mean Average Precision at $K$ (MAP@$K$) metric for information retrieval (Schutze et al., 2008). Differently from vanilla MAP@$K$, in RankC $K$ varies across queries. The value $K$ for $q_i$ equals $N_i$, the number of its candidates. Given languages $l$ and $l'$, the consistency score between the two languages is defined as the mean value for the consistency of all translated query pairs $(q_i, q'_i) \in (Q_l, Q_{l'})$:

$$\text{RankC}(l, l') = \frac{\sum_{i=1}^{|Q_l|} \text{consist}(q_i, q'_i)}{|Q_l|} \quad (3)$$

The consistency for each query pair is calculated by weighted averaging $P@j$ functions, which output the overlapping ratio among the candidates with top-$j$ highest probabilities[3]:

$$\text{consist}(q_i, q'_i) = \sum_{j=1}^{N_i} w_j * P@j$$
$$P@j = \frac{1}{j}|\{c_i^1 \dots c_i^j\} \cap \{c'^1_i \dots c'^j_i\}| \quad (4)$$

The weight $w_j$ for each $P@j$ is defined below.

**Ranking-based weights** Intuitively, candidates with a higher ranking should have more influence on the consistency score. To achieve this, RankC adopts a weighted average over all $P@j$s, where $P@j$ with a smaller $j$ is given a higher weight $w_j$ to emphasize the effect of candidates with high probabilities. However, the predicted probabilities cannot be directly used since they are different for the candidates of $q_i$ and $q'_i$. To solve this issue, we introduce normalized weights based on softmax over the value $j$:

$$w_j = \frac{e^{N_i - j}}{\sum_{k=1}^{N_i} e^{N_i - k}} \quad (5)$$

where $N_i$ is the number of candidates for queries $q_i$ and $q'_i$.[4] Combining equations 3, 4 and 5, the RankC metric becomes:

$$\text{RankC}(l, l') = \frac{\sum_{i=1}^{|Q_l|} \sum_{j=1}^{N_i} \frac{e^{N_i - j}}{\sum_{k=1}^{N_i} e^{N_i - k}} * P@j}{|Q_l|}$$
$$P@j = \frac{1}{j}|\{c_i^1 \dots c_i^j\} \cap \{c'^1_i \dots c'^j_i\}| \quad (6)$$

A RankC computation example is given in Appendix D and the interpretation of high/low RankC

| Property | BMLAMA-17 | BMLAMA-53 |
|---|---|---|
| # Languages | 17 | 53 |
| # Relations | 41 | 30 |
| # Queries | 6792*17 | 3070*53 |
| # Candidates (Avg) | 9.71 | 9.56 |

Table 1: Statistics of the Balanced Multilingual LAMA (BMLAMA) benchmark used in our analysis.

scores is discussed in Appendix E.

We performed an empirical comparison of RankC to the previously used metric (COverlap, cf. Eq. 1) on the same dataset. The results in Appendix F show that nearly all language pairs with a high COverlap score also receive a high RankC score. Additionally, RankC reveals some new high-consistency pairs that had a low COverlap score due to low probing accuracy.

## 4 Experimental Setup

**Datasets** As explained in Sect. 3.2, RankC requires that the queries and their candidates are translated into all evaluated languages. We therefore extract from X-FACTR (Jiang et al., 2020) and MLAMA (Kassner et al., 2021) all queries that fulfill this criterion. We call the resulting multi-parallel dataset Balanced Multilingual LAnguage Model Analysis (**BMLAMA**), and release it in two versions: BMLAMA-17 including 6.7k queries in 17 languages (close to X-FACTR which includes 23 languages), and BMLAMA-53 including 3k queries in 53 languages (same as MLAMA). The detailed statistics are shown in Table 1.

**Models** Previous work on multilingual knowledge probing (Jiang et al., 2020; Kassner et al., 2021) focused on encoder-only PLMs, such as mBERT (Devlin et al., 2019) or XLM-RoBERTa (Liu et al., 2019). However, since decoder-only PLMs have become mainstream in the current NLP era, our experiments also include the decoder-only BLOOM series (560m, 1.1b, 1.7b, 3b parameters) (Scao et al., 2022) and the encoder-decoder mT5-large (1.2b) (Xue et al., 2021), in addition to the encoder-only XLM-RoBERTa-large (354m).

---

[3]We recall that candidates $\{c_i^1, \dots, c_i^{N_i}\}$ have been sorted by their PLM probabilities from highest to lowest (cf. Sect. 3).

[4]For alternative ranking-based weighting schemes, see Appendix C.

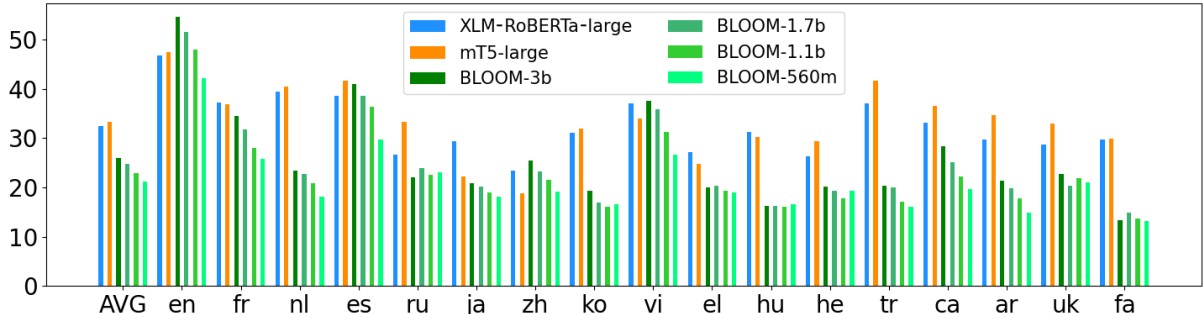

Figure 2: Factual knowledge probing accuracy (%) in various multilingual PLMs, measured on BMLAMA-17.

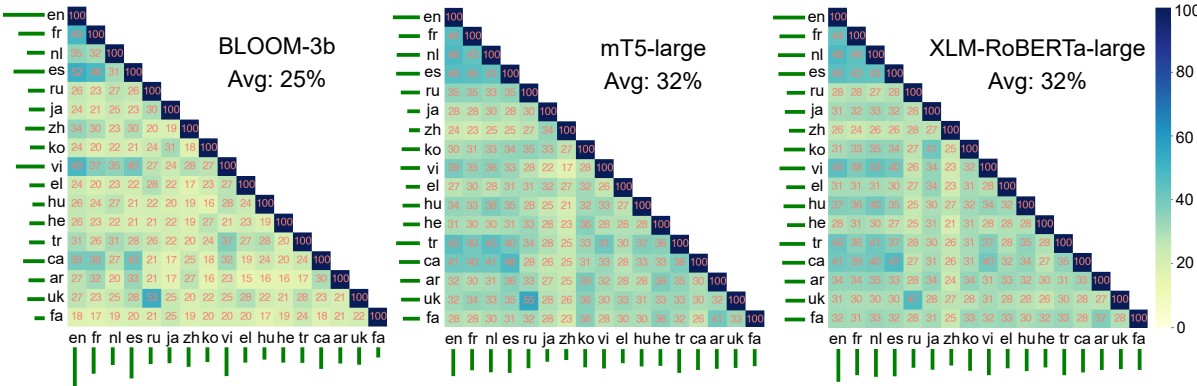

Figure 3: Knowledge consistency (RankC %) between language pairs in the PLMs, with darker shading denoting higher-consistency pairs. *Green* bars: Probing accuracy of each language.

## 5 Main Consistency Results

Before looking at consistency, we present in Figure 2 the factual probing accuracy results of the three PLMs on BMLAMA-17.[5] We first notice that the encoder-only XLM-RoBERTa-large and encoder-decoder mT5-large models outperform the whole decoder-only BLOOM series in terms of average probing accuracy. The trend across languages is similar among the three models, however, BLOOM stands out with an English accuracy that is much higher than all other languages. Regarding model size (BLOOM series, green bars), we find that increasing the number of parameters leads to slight but consistent improvements in factual probing accuracy, which is in line with previous work (Petroni et al., 2019).

Our XLM-RoBERTa-large results are consistent with those reported by Jiang et al. (2020) on X-FACTR, which demonstrates the reliability of our multi-parallel dataset BMLAMA.

### 5.1 Consistency in Different PLMs

Figure 3 presents RankC results for the three PLMs. The first observation is that average consistency[6] is rather low in all models, and lowest in BLOOM-3b (25%). This negative result is in line with the low overlap ratio of correct predictions observed by Jiang et al. (2020) on mBERT.

We now zoom in on the comparison among different language pairs, which is made possible by the new RankC metric and the balanced dataset BMLAMA. Here, we find that the European languages English, French, Dutch, Spanish, and Catalan share a considerable amount of knowledge in mT5-large and XLM-RoBERTa-large. A similar pattern applies to BLOOM-3b with the exception of Dutch, which is expected as this language was not included in this model's training corpus.[7] In addition, Vietnamese and Turkish achieve remarkable consistency with the above European languages in all PLMs. A common feature of these languages is that they all use the same script (Latin). Another notable high-consistency pair is that of Russian-

---

[5]The experiments in this section use BMLAMA-17 as it enables better visualization. Results for BMLAMA-53 are given in Appendix H and discussed in the analysis of Section 6.

[6]Pairs of a language to itself (e.g. en-en) have always a RankC of 100% and are excluded from the average.

[7]https://huggingface.co/bigscience/bloom-3b

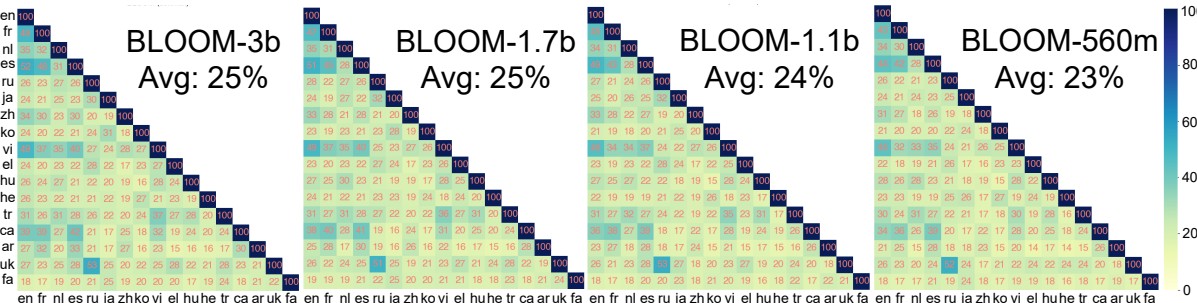

Figure 4: Cross-lingual consistency (RankC) shows little fluctuation among PLMs of different scales. The average probing accuracy of the four models is 25.97%, 24.77%, 22.93%, and 21.14%, respectively.

Ukrainian, two languages that use the same script (Cyrillic) and are also closely related.

These observations suggest that various language properties influence the CLC of multilingual knowledge. We examine a number of such properties in Section 6.1.

## 5.2 Effect of Model Size

As observed above (green bars in Figure 2) and by previous work (Petroni et al., 2019), the ability to retrieve correct knowledge grows with model size when other factors are fixed. We ask whether this is also the case for CLC. However, the BLOOM results in Figure 4 show only a minor variation in average RankC (+2%) when moving from the smallest to the largest model in our series, i.e. a 5-fold increase in parameters. While this pattern cannot be safely generalized to other models, it does suggest that cross-lingual consistency may remain a problem in very large-scale PLMs.[8]

## 6 Factors of Cross-Lingual Consistency

In this section, we examine the different degrees of CLC that are observed in different language pairs, and the factors that may explain such differences.

## 6.1 Typological Similarity

Typological features have been proven useful to model fine-grained similarities among languages and guide transfer learning techniques for a variety of multilingual NLP tasks (Ponti et al., 2019; Üstün et al., 2022). Could such features also explain some of the variance observed in the consistency of factual knowledge within multilingual PLMs? For example, we may expect that languages with similar grammar and word order, or with related vocabularies share a higher degree of linguistic representation.

tations within a multilingual model. We may also expect that languages spoken in the same world region have higher chances of encountering mentions of the same entities and events in the training data.

To answer this question, we obtain four types of typological similarity (syntactic, genetic, geographic, and phonological) from lang2vec (Littell et al., 2017), an open-source library that provides pre-computed similarities based on various typological databases.[9] Next, we compute the Pearson correlation coefficient (Cohen et al., 2009) between RankC scores and the typological similarity of all language pairs in BMLAMA.

Results are shown in Table 2 for both BMLAMA-17 and the smaller but more multilingual BMLAMA-53.[10] For BMLAMA-17, we find that RankC has a moderate correlation with genetic similarity, a weak correlation with geographical similarity, but no significant correlation with syntactic similarity. As expected, no correlation is observed with phonological similarity. The correlation results on the more comprehensive dataset BMLAMA-53 are similar except for syntactic similarity which acquires a weak positive correlation. Somewhat surprisingly, genetic and geographical similarities see their correlations slightly decreased in this larger dataset, which might be due to the presence of noise in the typological vectors of low-resource languages.

An important characteristic of genetically related languages is that they tend to have many words in

---

[8]See Appendix I for additional results on models with 7b parameters.

[9]The similarity score for a language pair ranges from 0 to 1, where 1 represents maximal similarity. For example, German and Polish have a geographical similarity of 1 because they are spoken in neighboring regions but a genetic similarity of only 0.14 because they belong to very different branches of the Indo-European family. Genetic similarity is based on the distance of two languages in the Glottolog tree (Hammarström et al., 2017).

[10]RankC and genetic similarity values on BMLAMA-53 are given in Appendix H.

| BMLAMA-17 | Syntactic | Genetic | Geographical | Phonological | Voc(BMLAMA) | Voc(Flores) |
|---|---|---|---|---|---|---|
| BLOOM-3b | 0.08 (4e-01) | **0.52** (8e-11) | 0.37 (8e-06) | -0.05 (6e-01) | **0.70** (2e-21) | **0.49** (2e-09) |
| mT5-large | 0.12 (2e-01) | **0.50** (4e-10) | 0.43 (2e-07) | 0.00 (1e-00) | **0.71** (2e-22) | **0.68** (5e-20) |
| RoBERTa | -0.04 (6e-01) | **0.42** (4e-07) | 0.34 (4e-05) | -0.09 (3e-01) | **0.80** (8e-32) | **0.79** (1e-30) |
| BMLAMA-53 | Syntactic | Genetic | Geographical | Phonological | Voc(BMLAMA) | Voc(Flores) |
| BLOOM-3b | 0.14 (2e-07) | **0.40** (2e-53) | 0.15 (1e-08) | 0.05 (6e-02) | **0.69** (3e-195) | **0.61** (2e-140) |
| mT5-large | 0.20 (2e-13) | **0.31** (2e-32) | 0.23 (3e-18) | -0.02 (4e-01) | **0.74** (8e-243) | **0.65** (2e-168) |
| RoBERTa | 0.22 (4e-16) | **0.28** (5e-26) | 0.21 (8e-15) | 0.00 (1e-00) | **0.70** (5e-205) | **0.63** (7e-156) |

Table 2: Pearson correlation (with p-value) between language-pairwise knowledge consistency (RankC) and various typological similarities (left) or with subword vocabulary overlap (right).

common or with a common ancestor. Thus, the moderate correlation of RankC with genetic similarity, combined with the weak correlation with syntactic and geographic similarity, suggests that vocabulary overlap may be a more important factor for CLC than having similar grammar and word order or being spoken in nearby regions.

## 6.2 Subword Vocabulary Overlap

Based on the above observations, we investigate whether a crude measure of vocabulary overlap could also be a good predictor of CLC. Specifically, we extract the vocabularies of strictly parallel corpora in our evaluated languages and measure their pairwise overlap:

$$\frac{|V(l) \cap V(l')|}{|V(l) \cup V(l')|} \in [0, 1] \qquad (7)$$

We consider two corpora: BMLAMA itself and Flores-200 (Costa-jussà et al., 2022). The former is expected to be very relevant, but the resulting correlation may be less generalizable as it is the same corpus on which consistency itself is measured. By contrast, the latter is a mix-domain set of 2k sentences translated from English into 200 languages for the purpose of MT evaluation. Because we are interested in the extent to which different languages make use of the exact same word representations, we segment the corpora with the model's tokenizer *before* measuring vocabulary overlap, which makes this metric model-dependent.[11]

Shown in Table 2 (right), the Pearson correlation scores on BMLAMA prove that subword vocabulary overlap has a significantly strong impact on the cross-lingual consistency of knowledge in the PLMs, overshadowing the effect of genetic

[11] See Appendix H for the vocabulary overlap scores of all language pairs in BMLAMA-53.

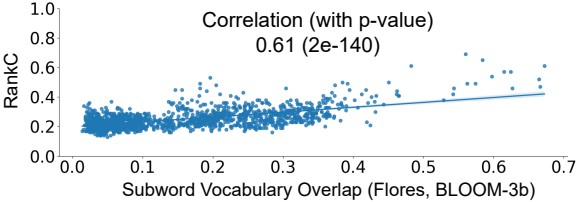

Figure 5: Linear regression between subword vocabulary overlap (Flores) and CLC measured on BMLAMA-53 for BLOOM-3b. For results on mT5-large and XLM-RoBERTa-large, see Appendix H.

similarity. This suggests that factual knowledge may be primarily percolating across languages in a rather shallow way (through the shared use of some subword embeddings) and conversely, it may be hampered in the absence of such anchors even when languages are related. For instance, the highest-consistency pair in BLOOM-3b is Ukrainian-Russian, which lies close in the language tree (genetic similarity: 0.8) and shares a vast number of subword vocabulary overall (vocabulary overlap: 0.76). However, when querying the working place of David Cameron, BLOOM-3b predicts the correct answer in the Russian query ('London') but a wrong answer in Ukrainian ('Moscow'). This suggests that the correct knowledge did not transfer from Russian to Ukrainian due to the limited subword overlap between these two queries (0.17). When vocabulary overlap is measured on Flores (last column of Table 2), correlation is lower but still significantly positive, indicating our findings are not limited to our benchmark. The correlation between cross-lingual knowledge consistency and vocabulary overlap is clearly illustrated in Figure 5.

The strong dependence of CLC on shallow vocabulary overlap partly explains why increasing

the model size does not have a positive effect (cf. Section 5.2). We speculate that larger subword vocabularies may actually lead to *lower* consistency as the chances of sharing parts of words between any two languages decrease. We leave a further investigation of this hypothesis to future work.

## 7 Case Study: Cross-Lingual Consistency and Knowledge Incorporation

Previous work (Jiang et al., 2020; Kassner et al., 2021; Artetxe et al., 2022) and our probing results indicate the amount of knowledge in low-resource languages is limited. Simply training a new PLM on larger non-English corpora is time-consuming and the cost is not affordable by most universities and other research institutions (Ding et al., 2022). A promising solution is to incorporate external knowledge through fine-tuning methods (Hu et al., 2022) or directly editing the PLM's weights in a very targeted way (De Cao et al., 2021; Meng et al., 2022). To make the process feasible in multilingual scenarios and avoid unexpected effects, it is important to understand whether and how inserting a piece of knowledge in one language affects other languages within the PLM, including the most and least susceptible languages. In this section, we conduct a first case study on this problem and its interplay with CLC.

**Rank-One Model Editing (ROME)**  Proposed by Meng et al. (2022), this state-of-the-art model editing technique based on neuron interpretability outperforms several other editing techniques both in terms of specificity and generalization. In short, this technique directly modifies weights in the early feed-forward layers of the PLM, where a factual association has been located via causal interventions.

**Counterfactual knowledge**  Following Meng et al. (2022), we consider the task of inserting counterfactual knowledge into the PLM, such as the factually wrong *'Steve Jobs worked for Microsoft'*. Because such factual associations were never observed during pre-training, this approach avoids the risk of inserting facts that were already considered likely by the model.

**Case study**  We examine BLOOM-3b since ROME is currently only applicable to decoder-only models. English is chosen as the source language in which the fact is inserted. As target languages, we choose two that have high consistency (RankC) with English (Spanish and Vietnamese) and two

| Lang | RankC with en | Pre-editing Correct | Pre-editing Wrong | Post-editing Correct | Post-editing Wrong |
|------|---------------|---------|-------|---------|-------|
| \multicolumn{6}{c}{Steve Jobs worked for **Apple**/**Microsoft**} |
| en | - | **0.95** | 0.05 | 0.19 | **0.81** |
| es | 52 (high) | **0.93** | 0.07 | 0.12 | **0.88** |
| vi | 49 (high) | **0.99** | 0.01 | 0.24 | **0.76** |
| hu | 26 (low) | **0.95** | 0.05 | **0.81** | 0.19 |
| el | 24 (low) | **0.99** | 0.01 | **0.91** | 0.09 |
| \multicolumn{6}{c}{IBM Connections is created by **IBM**/**Adobe**} |
| en | - | **0.93** | 0.07 | 0.38 | **0.63** |
| es | 52 (high) | **0.96** | 0.04 | 0.36 | **0.64** |
| vi | 49 (high) | **0.96** | 0.04 | 0.31 | **0.69** |
| hu | 26 (low) | **0.99** | 0.01 | **0.85** | 0.15 |
| el | 24 (low) | **0.99** | 0.01 | **0.68** | 0.32 |
| \multicolumn{6}{c}{Sandy Bridge was a product of **Intel**/**Samsung**} |
| en | - | **0.99** | 0.01 | 0.40 | **0.60** |
| es | 52 (high) | **0.98** | 0.02 | 0.22 | **0.78** |
| vi | 49 (high) | **0.99** | 0.01 | 0.09 | **0.91** |
| hu | 26 (low) | **0.93** | 0.07 | **0.60** | 0.40 |
| el | 24 (low) | **0.93** | 0.07 | **0.55** | 0.45 |

Table 3: Normalized logits for predicting the candidates before and after model editing BLOOM-3b in English.

with low RankC (Hungarian and Greek). These languages are also diverse in terms of script and relatedness to English. Six queries are picked by ensuring that the PLM selects the originally correct answer as highly probable before editing. We also ensure that, for each edited knowledge, the subject and object entities are the same tokens across all languages. This eliminates the concern that, e.g., Spanish and Vietnamese get consistent predictions with English simply because of the lexical co-occurrence of their subject and object tokens in the evaluated queries. For the evaluation, we follow the setup in Meng et al. (2022) and narrow down candidate sets into two words – one correct and one wrong. The latter is the editing target of ROME. Fed with each query, the PLM calculates logit values for the correct and wrong answers, namely $\text{logit}_C$ and $\text{logit}_W$ respectively. These logits vary vastly among different languages. To focus on the relation between the original and edited fact, we normalize the logits following the previous work (Sarti et al., 2023) as $\frac{\text{logit}_C}{\text{logit}_C + \text{logit}_W}$ and $\frac{\text{logit}_W}{\text{logit}_C + \text{logit}_W}$. [12]

---

[12] The raw probability values are provided in Appendix K.

Table 3 shows the results for three queries. A very clear pattern emerges: when a fact is inserted in English, it propagates consistently to the high-CLC languages (i.e. Spanish and Vietnamese). Conversely, the low-CLC languages (Hungarian and Greek) are much less affected and still output higher probabilities for the correct answers even after model editing. The three remaining queries given in Appendix J show the same pattern.

Despite the small size of our study, the results suggest that CLC is not only a by-product of the existing knowledge in PLMs, but also represents the susceptibility to perturbation of a language when incorporating new knowledge in other languages. We see this as a promising direction to enhance the benefits of model editing in multilingual scenarios.

## 8 Conclusion

We analyzed the cross-lingual consistency (CLC) of factual knowledge in multilingual large-scale PLMs. We proposed a new metric, RankC, to quantify consistency independently from accuracy, and applied it to a cross-lingually balanced factual knowledge benchmark. Our comprehensive analysis shows that (i) average CLC is low across different PLMs and not visibly affected by model size; (ii) CLC of different language pairs within a PLM correlates significantly with genetic similarity, but the correlation with vocabulary overlap is notably stronger; (iii) novel facts inserted in language X via model editing are more likely to propagate to languages having higher CLC scores with X.

Taken together, our findings suggest that factual knowledge may primarily propagate across languages in a shallow way, that is, through shared word embeddings rather than through deeper language-agnostic representations. While this result has negative implications on the generalization abilities of modern PLMs, it could also be turned into an advantage to design efficient methods for knowledge incorporation in multilingual scenarios.

## Acknowledgements

This publication is part of the project LESSEN with project number NWA.1389.20.183 of the research program NWA-ORC 2020/21 which is (partly) financed by the Dutch Research Council (NWO). We also gratefully acknowledge the computational resources provided by the High-Performance Computing cluster Hábrók. RF is supported by the European Research Council (ERC) under the European Union's Horizon 2020 research and innovation programme (grant agreement No. 819455).

## Limitations

Due to restriction of our GPU resources, we could not test models larger than BLOOM-7.1b. Extending our analysis to larger-scale models in future work is encouraged to see if the same conclusions reached. Nevertheless, results in Figure 4 indicate that the average CLC grows extremely slowly with the increment of model scale.

The facts included in BMLAMA, while supposed to be universals, are likely to be more relevant to the Western world, which can introduce a bias in the evaluation. We inherit this problem from the benchmarks BMLAMA is built upon. Fixing this issue is not trivial, especially in comparative work where probing the exact set of facts across languages is a requirement, and should be given attention in future work.

## Ethics Statement

Since BMLAMA data is derived from previous works X-FACTR and MLAMA, queries in BMLAMA are also likely to encounter gender and racial bias issues, which are inevitable in the source Wikidata. However, this paper mostly focuses on the consistency between knowledge in different languages rather than the specific knowledge in a single language.

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

## A  GenBench Evaluation Card

Figure 4 shows the evaluation card for our experiments on BMLAMA according to the generalization taxonomy of Hupkes et al. (2022).

| **Motivation** | | | |
|---|---|---|---|
| *Practical* | *Cognitive* | *Intrinsic* | *Fairness* |
| ☐ | | | ☐ |
| **Generalisation type** | | | | | |
| *Compositional* | *Structural* | *Cross Task* | *Cross Language* | *Cross Domain* | *Robustness* |
| | | | ☑ | | |
| **Shift type** | | | |
| *Covariate* | *Label* | *Full* | *Assumed* |
| ☐ | | | |
| **Shift source** | | | |
| *Naturally occuring* | *Partitioned natural* | *Generated shift* | *Fully generated* |
| ☐ | | | |
| **Shift locus** | | | |
| *Train–test* | *Finetune train–test* | *Pretrain–train* | *Pretrain–test* |
| | | | ☐ |

Table 4: Evaluation card for experiments on BMLAMA.

## B  Prompt-based Probing

Given a query $q$ which carries the subject word and the mask slot(s), the ranking score of each word in the candidate sets is calculated by the following prompt-based probing methods (Kassner et al., 2021; Yin et al., 2022).

**Encoder-only** Fed into the PLM $M$, each candidate word $c$ is tokenized into multiple sub-words $s_1, \ldots, s_n$. For instance, 'chopsticks' is split as 'chop', 'stick', and 's' in BERT. The ranking score of each candidate $c$ is the average log probability of each sub-word occurring at their corresponding mask slot:

$$score_{enc}(c) = \frac{1}{n} \sum_{i=1}^{n} \log P_M([MASK]_i = s_i |$$
$$[MASK]_{<i} = s_{<i}, q) \quad (8)$$

**Encoder-Decoder** Given one mask slot, encoder-decoder PLMs are able to output a sequence of words. Therefore, the ranking score for each word $c$ in the candidate sets is represented as

$$score_{encdec}(c) = \log P_M([MASK] = c|q)$$
$$= \frac{1}{n} \sum_{i=1}^{n} \log P_M(s_i|s_{<i}, q) \quad (9)$$

**Decoder-only** Decoder-only PLMs abolish the use of mask slots in their pre-training and only perform next-token predictions. Therefore, the ranking score of each word $c$ is defined as the output probability of the sub-word sequence of $q(c)$, where the candidate word $c$ is inserted into the mask slot of $q$. Formally, the score is defined as

$$score_{dec}(c) = \frac{1}{l} \sum_{i=1}^{l} \log P_M(q(c)_i|q(c)_{<i}) \quad (10)$$

where $l$ is the length of sub-word sequence of $q(c)$.

## C  Ranking-Based Weight

RankC uses a weighted average to give more importance to candidates with a higher ranking (Eq. 3):

$$\text{RankC}(l, l') = \frac{\sum_{i=1}^{|Q_l|} \text{consist}(q_i, q_i')}{|Q_l|}$$

In alternative to the softmax-normalized weights $w_j$ introduced in Eq. 5, we also considered weights based on 1-norm:

$$w_j = \frac{N_i - j}{\sum_{k=1}^{N_i}(N_i - k)} \quad (11)$$

or 2-norm:

$$w_j = \frac{(N_i - j)^2}{\sum_{k=1}^{N_i}(N_i - k)^2} \qquad (12)$$

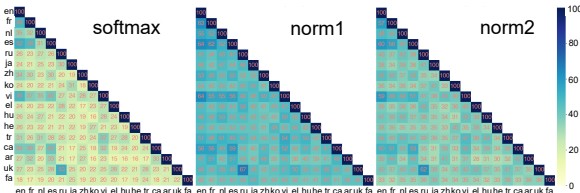

Figure 6: Comparison of different ranking-based weighting schemes on BLAMA-17, for BLOOM-3b.

Experimental results in Figure 6 illustrate that the softmax weighting scheme achieves the best visualization by highlighting the most consistent pairs. Therefore, we selected it as RankC weighting scheme in all our experiments.

## D  RankC Computation Example

To get a clearer view of the RankC metric, we provide an example of computing the score between the English-Spanish pair. For simplicity, the number of candidates is narrowed to 3 for each query.

When probed with the original language of The Godfather in English ($q_i$) and Spanish ($q_i'$), the PLM predicts the ranking scores for the candidates of the two queries. Based on the scores, the candidate sets are sorted as {Italian, English, Russian} for English and {italiano, ruso, inglés} for Spanish.

$P@1$ of these two sorted candidate sets is $\frac{1}{1}$ since although incorrect, the answers with the highest probabilities are consistent. The weight $w_1$ for it is calculated as $\frac{e^2}{e^0+e^1+e^2} = 0.67$, which highlights the effect of the top-1 predictions.

The value for $P@2$ is $\frac{1}{2}$ since 'Italian' is the only consistent answer appearing in top-2 predictions of both sets. The weight $w_2$ is $\frac{e^1}{e^0+e^1+e^2} = 0.24$.

The value for $P@3$ is $\frac{3}{3}$ because (translations of) the first 3 elements in the English candidate set are all covered by the first 3 elements in the Spanish candidate set. However, it is multiplied by a low weight $w_3$, equaling $\frac{e^0}{e^0+e^1+e^2} = 0.09$. In practice, the average length of candidate sets in BMLAMA is around 10. Given $j \in \{1, \ldots, 10\}$, the weights for $P@j(j \geq 6)$ are lower than 0.004.

The consistency between $q_i$ and $q_i'$ is then:

$$\begin{aligned}
\mathrm{consis}(q_i, q_i') &= \sum_{j=1}^{N_i} w_j * P@j \\
&= 0.67 * \frac{1}{1} + 0.24 * \frac{1}{2} + 0.09 * \frac{3}{3} \\
&= 0.88
\end{aligned}$$

Averaging the consistency between all translated query pairs, the CLC between English and Spanish is represented as the RankC score.

$$\mathrm{RankC}(l, l') = \frac{\sum_{i=1}^{|Q_l|} \mathrm{consist}(q_i, q_i')}{|Q_l|} \qquad (13)$$

## E  Range of RankC Values

In practice, if a model outputs the same answers (the ones with the highest probabilities) for all queries in two languages, the $P@1$ values for each query are constantly 1 and the RankC value between these two languages should be equal or larger than $\frac{\sum_{i=1}^{|Q_l|} w_1 * P@1}{|Q_l|} = \frac{\sum_{i=1}^{|Q_l|} w_1}{|Q_l|}$, where $w_1$ is the weight for the candidates with the highest probability, which varies with the length of the candidate sets. However, since the average candidate is about 10 for BMLAMA, $w_1$ can be approximated by $\frac{e^9}{e^0 + \ldots + e^9}$ for nearly all queries. The RankC score under this 'all-same-answer' hypothesis is then

$$\mathrm{RankC}(l, l') \approx \frac{|Q_l| * \frac{e^9}{e^0 + \ldots + e^9}}{|Q_l|} = 63\%.$$

Therefore, two languages are considered highly consistent if their average RankC score is around 63% or larger.

## F  Comparison of RankC and COverlap

Figure 7 compares the proposed RankC scores to COverlap scores Jiang et al. (2020), both computed on BMLAMA-17. The pairs with consistency greater or equal to the average of all language pairs in each PLM are considered high-consistency pairs and marked with squares. We find that nearly all high-consistency pairs found by the previous metric are successfully retrieved by our RankC metric. Despite the three missing pairs, RankC mines a total of 7 pairs that are underestimated by OnlyCorrect, where one of the languages has low probing accuracy. This demonstrates the effectiveness of the RankC metric at disentangling consistency from accuracy.

| Model | CLC | Acc | en | fr | nl | es | ru | ja | zh | ko | vi | el | hu | he | tr | ca | ar | uk | fa |
|---|---|---|---|---|---|---|---|---|---|---|---|---|---|---|---|---|---|---|---|---|
| RoBERTa | 32.0 | 32.5 | 46.8 | 37.3 | 39.4 | 38.6 | 26.7 | 29.4 | 23.4 | 31.1 | 37.1 | 27.3 | 31.3 | 26.3 | 37.0 | 33.2 | 29.7 | 28.7 | 29.8 |
| mT5-large | 32.3 | 33.4 | 47.4 | 36.8 | 40.5 | 41.6 | 33.4 | 22.2 | 18.9 | 32.0 | 34.1 | 24.8 | 30.3 | 29.5 | 41.8 | 36.6 | 34.6 | 32.9 | 30.0 |
| LLaMA-7b | 28.5 | 28.5 | 51.7 | 31.3 | 41.5 | 38.6 | 34.5 | 19.9 | 15.2 | 19.0 | 22.3 | 20.4 | 28.4 | 23.5 | 23.2 | 34.3 | 23.1 | 35.5 | 22.3 |
| BLOOM-7.1b | 26.3 | 27.8 | 56.4 | 35.7 | 27.0 | 45.3 | 22.3 | 20.4 | 28.6 | 18.2 | 41.9 | 20.4 | 16.8 | 23.3 | 21.1 | 33.1 | 24.7 | 22.6 | 15.2 |
| BLOOMZ-3b | 24.3 | 23.9 | 52.3 | 37.5 | 20.6 | 39.8 | 16.5 | 18.6 | 24.0 | 17.2 | 37.6 | 15.5 | 16.9 | 16.1 | 16.5 | 24.0 | 21.3 | 18.2 | 13.2 |
| BLOOM-3b | 25.2 | 26.0 | 54.7 | 34.6 | 23.4 | 41.0 | 22.1 | 20.9 | 25.4 | 19.4 | 37.7 | 20.1 | 16.2 | 20.2 | 20.3 | 28.3 | 21.4 | 22.7 | 13.3 |
| BLOOM-1.7b | 24.8 | 24.8 | 51.5 | 31.8 | 22.7 | 38.7 | 24.0 | 20.1 | 23.3 | 17.0 | 35.9 | 20.3 | 16.2 | 19.3 | 20.0 | 25.1 | 19.9 | 20.4 | 14.8 |
| BLOOM-1.1b | 23.9 | 22.9 | 48.1 | 28.0 | 20.9 | 36.5 | 22.6 | 18.9 | 21.6 | 16.1 | 31.3 | 19.3 | 16.1 | 17.7 | 17.1 | 22.2 | 17.8 | 21.9 | 13.8 |
| BLOOM-560m | 23.1 | 21.1 | 42.2 | 25.9 | 18.1 | 29.7 | 23.1 | 18.1 | 19.1 | 16.6 | 26.7 | 19.1 | 16.6 | 19.4 | 16.0 | 19.6 | 14.9 | 21.0 | 13.2 |

Table 5: Overall probing results. Probing accuracy rises with the increment of model size in BLOOM series.

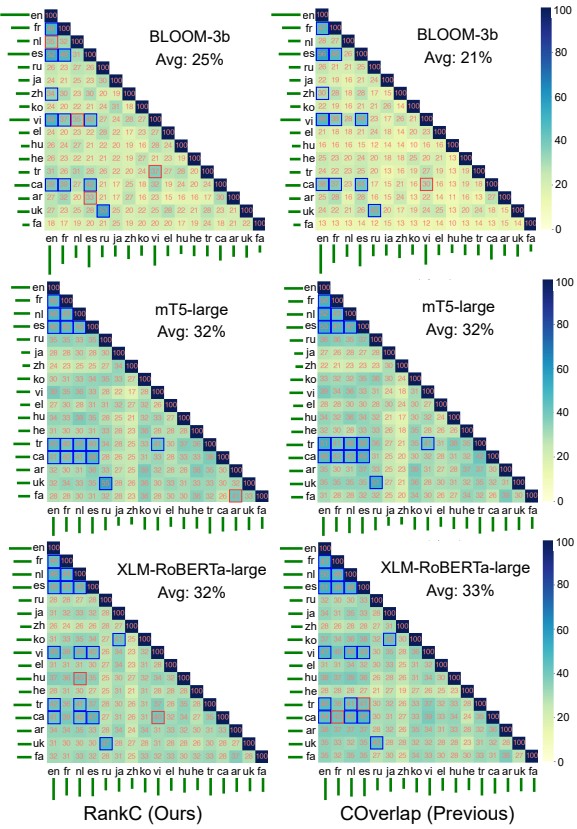

Figure 7: Comparison between RankC and COverlap on BLOOM-3b, mT5-large, and XLM-RoBERTa-large. *Blue* squares: High-consistency pairs retrieved by both metrics. *Red* squares: High-consistency pairs mined by only one metric. *Green* bars: Probing accuracy.

| Dataset | Voc(BMLAMA) | Voc(Flores) |
|---|---|---|
| BMLAMA-17 | **0.83** (3e-06) | **0.79** (2e-05) |
| BMLAMA-53 | **0.76** (4e-18) | **0.75** (6e-18) |

Table 6: Pearson correlation (with p-value) between language-pairwise knowledge consistency (RankC) and vocabulary overlapping when only considering the supported languages of BLOOM-3b.

all BMLAMA languages in our main CLC evaluation. As suggested by previous work (Blevins and Zettlemoyer, 2022), this could be due to the actual presence in the training data of some non-officially supported languages.

To ensure the presence of non-supported languages did not affect our analysis of BLOOM, we further calculate the correlation scores between CLC and word-overlapping when only considering the supported languages. As shown in Table 6, the correlation scores between CLC and sub-word vocabulary overlap become even higher, confirming our finding that such overlap is a key predictor of consistency across languages.

## H  Additional Results on BMLAMA-53

Figure 8 shows the cross-lingual consistency of language pairs in BMLAMA-53, on BLOOM-3b, mT5-large, and XLM-RoBERTa-large. Overall, the average consistency is in line with the results on BMLAMA-17, with only a 2% decrease on XLM-RoBERTa-large due to the increasing number of languages.

Figure 10 shows the pairwise genetic similarity scores in BMLAMA-53. For instance, the Russian-Ukrainian pair here achieves 80% genetic similarity while it also shows high CLC in all PLMs.

## G  Effect of Non-Supported Languages in BLOOM

BLOOM does not officially support all languages in BMLAMA. Despite that, we found the factual knowledge probing accuracy of the non-supported languages to be considerably higher than randomly guessing (cf. Table 5) and thus decided to include

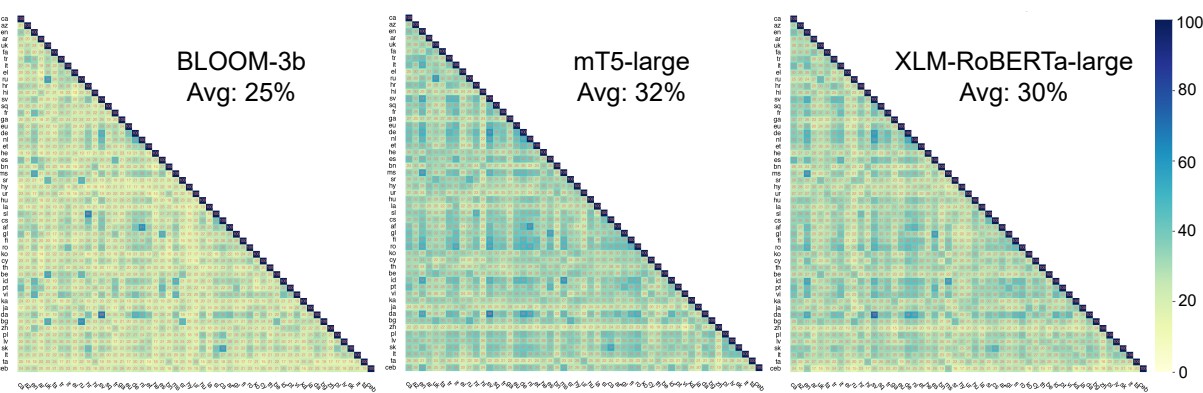

Figure 8: RankC scores between language pairs in BMLAMA-53.

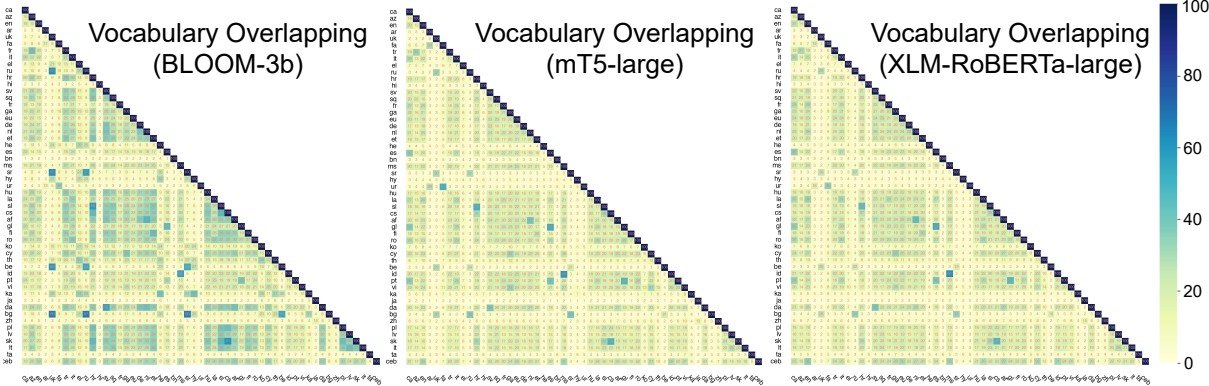

Figure 9: Subword overlapping for languages in BMLAMA-53, varying between PLMs due to different tokenizers.

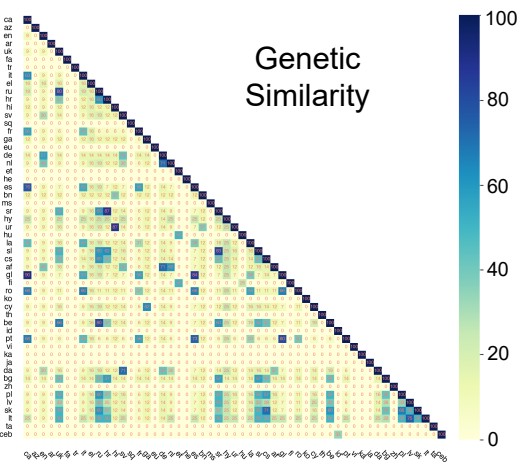

Figure 10: Genetic similarity between language pairs in BMLAMA-53, calculated by lang2vec (Littell et al., 2017) based on the distance of two languages in the Glottolog tree (Hammarström et al., 2017).

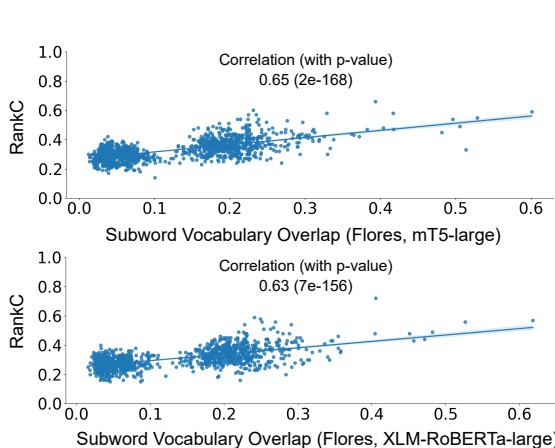

Figure 11: Linear regression between subword vocabulary overlap (Flores) and CLC measured on BMLAMA-53 for mT5-large and XLM-Roberta-large.

Figure 9 shows the subword vocabulary overlap scores in BMLAMA-53. We observe that high subword vocabulary overlapping leads to high RankC scores, such as sr-uk/be-uk/bg-uk/bg-sr/sk-sc pairs in BLOOM-3b, sl-hr/sv-da/id-ms pairs in

mT5-large, and gl-es/id-ms/pt-gl pairs in XLM-RoBERTa-large.

Figure 11 shows the correlation between subword vocabulary overlap (Flores) and CLC (RankC) computed with BMLAMA-53 for mT5-large and XLM-RoBERTa-large.

## I  Knowledge Consistency on More Models

We conducted additional experiments on more models than those included in the main body: namely, BLOOM-7.1b, a larger model in the BLOOM series; BLOOMZ-3b (Muennighoff et al., 2023), a branch of BLOOM-3b fine-tuned with human instruction; LLaMA-7b (Touvron et al., 2023), one of the cutting-edge PLMs trained on much more tokens (i.e. 1.4T) than BLOOM (i.e. 366B). Average CLC and probing accuracy for all PLMs are presented in Table 5.

BLOOM-7.1b: Probing accuracy raises 1.8% relative to BLOOM-3b, while the average CLC still grows relatively slowly (1.1%) in the BLOOM series.

BLOOMZ-3b: both average CLC and probing accuracy are lower than BLOOM-3b, suggesting human instruction fine-tuning has a mildly negative effect on factual knowledge. Nevertheless, average cross-lingual consistency (CLC) still remains less affected, and the heatmaps are nearly the same as BLOOM-3b since they use the exact same tokenizer. This further strengthens our conclusion that CLC is a property independent of accuracy and mostly relies on the subword overlapping.

LLaMA-7b: Probing accuracy only slightly outperforms BLOOM-7.1b having a similar number of parameters. For average CLC, LLaMA-7b reaches 28.5% while BLOOM-7.1b reaches 26.3%. In addition, we calculate the Pearson correlation scores between CLC and vocabulary overlapping of these two models. Results in Table 7 also support our finding that CLC positively correlates with the overlapping of sub-word vocabulary.

| BMLAMA-17 | Voc(BMLAMA) | Voc(Flores) |
|---|---|---|
| LLaMA-7b | **0.60** (2e-14) | **0.52** (1e-10) |
| BLOOM-7.1b | **0.69** (4e-20) | **0.48** (3e-09) |

Table 7: Pearson correlation (with p-value) between language-pairwise knowledge consistency (RankC) and vocabulary overlapping on LLaMA-7b and BLOOM-7.1b.

## J  Additional Cases of Counterfactual Knowledge Incorporation

| Lang | RankC with en | Pre-editing Correct | Wrong | Post-editing Correct | Wrong |
|---|---|---|---|---|---|
| Steve Jobs, who was employed by **Apple**/**Microsoft** | | | | | |
| en | - | **0.93** | 0.07 | 0.37 | **0.63** |
| es | 52 (high) | **0.93** | 0.07 | 0.34 | **0.66** |
| vi | 49 (high) | **0.91** | 0.09 | 0.40 | **0.60** |
| hu | 26 (low) | **0.82** | 0.18 | **0.75** | 0.25 |
| el | 24 (low) | **0.99** | 0.01 | **0.97** | 0.03 |
| Elon Musk worked for **Tesla**/**Chanel** | | | | | |
| en | - | **0.99** | 0.01 | 0.43 | **0.57** |
| es | 52 (high) | **0.99** | 0.01 | 0.18 | **0.82** |
| vi | 49 (high) | **0.99** | 0.01 | 0.40 | **0.60** |
| hu | 26 (low) | **0.98** | 0.02 | **0.58** | 0.42 |
| el | 24 (low) | **0.96** | 0.04 | **0.95** | 0.05 |
| Sandy Bridge is created by **Intel**/**Samsung** | | | | | |
| en | - | **0.99** | 0.01 | 0.33 | **0.67** |
| es | 52 (high) | **0.99** | 0.01 | 0.20 | **0.80** |
| vi | 49 (high) | **0.99** | 0.01 | 0.14 | **0.86** |
| hu | 26 (low) | **0.98** | 0.02 | **0.55** | 0.45 |
| el | 24 (low) | **0.89** | 0.11 | **0.86** | 0.14 |

Table 8: Normalized logits for predicting the candidates before and after model editing BLOOM-3b in English, for three additional queries.

Three extra queries are shown in Table 8. The languages with high RankC scores with English (i.e. es, vi) also change synchronously when incorporating new knowledge into BLOOM-3b. Meanwhile, the languages with low RankC scores are less affected, in line with our findings in Section 7.

Note that the first query ('*Steve Jobs, who was employed by __*') is a paraphrase of the first query shown in the main body (Table 3). The similar result obtained with the two paraphrases suggests that the specific prompt has little influence on the effect of model editing on different languages (Gao et al., 2021). Similar results are also observed in the two cases where the relation is fixed ('*__ worked for __*') and the model is queried about different subject entities (i.e. *Steve Jobs* in Table 3 and *Elon Musk* in Table 8).

## K  Raw Logits Before/After Model Editing

For replicability, Table K shows the raw (unnormalized) logits of the model editing case study.

| Lang | RankC with en | Pre-editing | | Post-editing | |
|---|---|---|---|---|---|
| | | Correct | Wrong | Correct | Wrong |
| colspan="6" | Steve Jobs worked for **Apple**/**Microsoft** |
| en | - | **4.1e-1** | 2.2e-2 | 1.4e-3 | **6.1e-3** |
| es | 52 (high) | **4.0e-1** | 2.9e-2 | 1.2e-3 | **9.0e-3** |
| vi | 49 (high) | **6.9e-1** | 9.0e-3 | 1.3e-3 | **4.1e-3** |
| hu | 26 (low) | **8.1e-5** | 4.1e-6 | **2.2e-5** | 5.3e-6 |
| el | 24 (low) | **8.4e-3** | 5.1e-5 | **6.0e-4** | 5.9e-5 |
| colspan="6" | IBM Connections is created by **IBM**/**Adobe** |
| en | - | **1.6e-2** | 1.3e-3 | 3.0e-4 | **5.0e-4** |
| es | 52 (high) | **3.9e-2** | 1.7e-3 | 8.0e-4 | **1.4e-3** |
| vi | 49 (high) | **1.3e-1** | 5.1e-3 | 1.6e-3 | **3.6e-3** |
| hu | 26 (low) | **9.3e-2** | 4.0e-4 | **2.2e-3** | 4.0e-4 |
| el | 24 (low) | **7.3e-2** | 1.1e-3 | **3.2e-3** | 1.5e-3 |
| colspan="6" | Sandy Bridge was a product of **Intel**/**Samsung** |
| en | - | **2.0e-1** | 1.5e-3 | 2.0e-4 | **3.0e-4** |
| es | 52 (high) | **1.1e-1** | 2.5e-3 | 2.0e-4 | **7.0e-4** |
| vi | 49 (high) | **5.7e-1** | 1.6e-3 | 6.0e-4 | **6.0e-3** |
| hu | 26 (low) | **5.3e-6** | 4.0e-7 | **9.6e-7** | 6.4e-7 |
| el | 24 (low) | **2.6e-3** | 2.0e-4 | **9.0e-6** | 7.4e-6 |
| colspan="6" | Steve Jobs, who was employed by **Apple**/**Microsoft** |
| en | - | **3.9e-1** | 2.9e-2 | 9.0e-4 | **1.5e-3** |
| es | 52 (high) | **5.9e-1** | 4.7e-2 | 1.4e-2 | **2.8e-2** |
| vi | 49 (high) | **5.3e-1** | 5.4e-2 | 1.2e-2 | **1.8e-2** |
| hu | 26 (low) | **1.0e-4** | 2.2e-5 | **5.9e-5** | 1.9e-5 |
| el | 24 (low) | **2.3e-3** | 2.0e-5 | **2.0e-4** | 6.3e-6 |
| colspan="6" | Elon Musk worked for **Tesla**/**Chanel** |
| en | - | **1.0e-1** | 2.0e-6 | 3.0e-4 | **4.0e-4** |
| es | 52 (high) | **1.1e-1** | 4.9e-6 | 2.0e-4 | **9.0e-4** |
| vi | 49 (high) | **2.9e-1** | 1.0e-6 | 2.0e-4 | **3.0e-4** |
| hu | 26 (low) | **4.7e-6** | 8.9e-8 | **1.1e-6** | 8.3e-7 |
| el | 24 (low) | **1.7e-8** | 6.7e-10 | **1.3e-8** | 6.0e-10 |
| colspan="6" | Sandy Bridge is created by **Intel**/**Samsung** |
| en | - | **2.2e-1** | 1.3e-3 | 1.0e-4 | **2.0e-4** |
| es | 52 (high) | **6.4e-1** | 1.4e-3 | 2.0e-4 | **8.0e-4** |
| vi | 49 (high) | **4.0e-1** | 1.5e-3 | 2.0e-4 | **1.2e-3** |
| hu | 26 (low) | **3.0e-4** | 7.3e-6 | **1.8e-6** | 1.4e-6 |
| el | 24 (low) | **4.0e-5** | 4.9e-6 | **1.3e-5** | 2.1e-6 |

Table 9: Raw (unnormalized) logits for predicting the candidates before and after model editing BLOOM-3b in English, for all the examined queries.