# OpenReview forum: "Cross-Lingual Consistency of Factual Knowledge in Multilingual Language Models"
_EMNLP/2023/Conference — EMNLP 2023 Main_

### Official Review · Reviewer_tJxb · 2023-08-03

**Soundness:** 5

**Excitement:**

4: Strong: This paper deepens the understanding of some phenomenon or lowers the barriers to an existing research direction.

**Paper Topic And Main Contributions:**

In this paper, the authors quantify the level of consistent factual information shared among languages in three distinct models. They differentiate between the consistency and accuracy of the knowledge, assessing the amount of shared knowledge between different language pairs without considering its correctness. The study presents a new metric, RankC, allowing to measure knowledge consistency independently of accuracy. Additionally, a new benchmark dataset, BMLAMA, is introduced, which allows prompt-based factual knowledge probing across 53 languages, using an equal number of prompts across languages. In addition, the paper investigates the impact of model type, model size, typological similarity and vocabulary overlap on cross-lingual consistency of factual knowledge.

**Questions For The Authors:**

**A)** With regards to my comment in the previous section, is there a specific reason why experiments were conducted for BLOOM on all potential language pairs, even when the model does not officially support certain languages? How much would the results be impacted if only supported language pairs would be considered?

**B)** Would you consider incorporating a brief discussion on the interplay between consistency and accuracy? The concept of quantifying consistency independent of accuracy is intriguing. However, addressing the question, "Does consistency of knowledge across languages imply its accuracy?" could potentially enrich the context and further highlight the significance of your results.

**Reasons To Accept:**

While this study primarily focuses on the cross-lingual consistency of factual knowledge rather than its correctness, it could indirectly enhance the latter as well. This is because it does not just identify factual inaccuracies in a particular language, but also investigates the reasons behind them. The conclusions of this paper point out that merely incorporating knowledge in one language and presuming its effective transfer to other languages is insufficient. This widely-held belief in cross-lingual NLP is rigorously questioned in this paper, which should encourage further research on this topic.

**Reasons To Reject:**

BLOOM officially supports only a fraction of the languages in BMLAMA-17 and BMLAMA-53, specifically 7 out of 17 and approximately 15 out of 53 respectively. This may significantly influence the cross-lingual consistency results for BLOOM, as well as the subsequently derived conclusions.

**Reproducibility:**

4: Could mostly reproduce the results, but there may be some variation because of sample variance or minor variations in their interpretation of the protocol or method.

**Reviewer Confidence:**

4: Quite sure. I tried to check the important points carefully. It's unlikely, though conceivable, that I missed something that should affect my ratings.

**Typos Grammar Style And Presentation Improvements:**

“Bloom” $\rightarrow$ “BLOOM”

 “xlm-RoBERTa”  $\rightarrow$ “XLM-RoBERTa”

---

> ### Author Rebuttal · Authors · 2023-08-27
>
> Thanks for providing valuable feedback and suggestions!
>
> 1)Regarding your question about the official supported languages of BLOOM, we decided to include all languages because we found the factual knowledge probing accuracy of the non-supported languages to be considerably higher than randomly guessing (cf. Table 4 in the Appendix). This could be due to the actual presence in the training data of some non-officially supported languages, or to the LM’s ability of making correct factual knowledge predictions even for unseen languages (for instance, because named entities are recognized by the LM anyway).
>
> To fully answer your question, we conducted additional experiments by only considering the officially supported languages , as you suggested.
>
> The CLC scores in BLOOM-3b still vary vastly between different language pairs, from 0.23 (ar-vi) to 0.52 (en-es).
>
> en$\quad$100
>
> fr $\quad$ 49$\quad$100
>
> es$\quad$ 52$\quad$48$\quad$100
>
> zh$\quad$ 34$\quad$30$\quad$30$\quad$100
>
> vi $\quad$ 49$\quad$37$\quad$40$\quad$29$\quad$100
>
> ca $\quad$39$\quad$39$\quad$42$\quad$25$\quad$32$\quad$100
>
> ar$\quad$ 27$\quad$32$\quad$33$\quad$27$\quad$23$\quad$30$\quad$100
>
> $\qquad$en$\quad$ fr$\quad$ es$\quad$zh$\quad$ vi$\quad$ ca$\quad$ ar
>
> Meanwhile, the correlation scores between CLC and sub-word vocabulary overlap (table below) become even higher, confirming our finding  that such overlap is a key predictor of consistency across languages. We will add these results to our paper:
>
> ——————————————————————
>
> | $\quad$ $\quad$ $\quad$ $\quad$ $ $ | Voc (BMLAMA) | Voc (Flores) |
>
> | BMLAMA-17 | $ $ $ $ $ $ 0.83 (3e-06) $ $ $ $ $ $ | 0.79 (2e-05) |
>
> | BMLAMA-53 | $ $ $ $ $ $ 0.76 (4e-18) $ $ $ $ $ $ | 0.75 (6e-18) |
>
> ——————————————————————
>
> 2)We agree that the interplay between consistency and accuracy is an interesting question that can be discussed more openly in the paper.
>
> According to our proposed accuracy-agnostic metric, there is not a direct connection between CLC and accuracy. However, if a model outputs correct answers (the ones with the highest probabilities) for all queries in two languages, the P@1 values for each query are constantly 1 and the CLC between these two languages should be equal or larger than $(\sum{w_1*P@1})/|Query| = \sum{w_1}/|Query|$, where $w_{1}$ is the weight for the candidate with the highest probability. Since there are ~10 candidates for nearly all queries, $w_1$ can be approximated as $e^9/(e^0+e^1+e^2+e^3+e^4+e^5+e^6+e^7+e^8+e^9)$, and the CLC score under this 'all-correct' hypothesis is $\sum{w_1}/|Query| \approx 63$%.
>
> The other situation where CLC can reach 63% is that the $\textbf{same}$ wrong predictions are made for all incorrectly predicted queries, which is very unlikely in practice.
>
> Therefore, this number can be seen as a threshold. If CLC between two languages reaches 63% on BMLAMA, they are very likely to make the $\textbf{same}$ and correct predictions for most queries.

---

### Official Review · Reviewer_xdFB · 2023-08-03

**Soundness:** 4

**Excitement:**

3: Ambivalent: It has merits (e.g., it reports state-of-the-art results, the idea is nice), but there are key weaknesses (e.g., it describes incremental work), and it can significantly benefit from another round of revision. However, I won't object to accepting it if my co-reviewers champion it.

**Paper Topic And Main Contributions:**

This paper studies the cross-lingual consistency of factual knowledge in multi-lingual pre-trained models. The authors propose a new metric to quantify the cross-lingual consistency (named RankC). To construct the benchmark datasets for probing multi-lingual pre-trained models with the same number of queries across different languages, they also modified existing X-FACTR and MLAMA to BMLAMA-17 and BMLAMA-53, respectively. The conclusion of paper includes (1) cross-lingual consistency is at a low level for existing mainstream multi-lingual pre-trained models (XLM-RoBERTa, mT5 and BLOOM); (2) the degree of cross-lingual consistency does not increase with the increase of parameters; (3) cross-lingual consistency of a language pair is relevant to the genetic similarity; (4) the influence of model editing in one language may transfer to other languages which have a high-level cross-lingual consistency with it.

**Reasons To Accept:**

- The motivation is clear and interesting. Cross-lingual consistency of factual knowledge seems important for understanding the implicit knowledge in multi-lingual pre-trained models.
- Well-organized and Well-written paper
- The proposed new metric could promote the research on this topic. The predicted probabilities are also considered in the metric, making it reasonable.

**Reasons To Reject:**

- I think the authors exaggerate their dataset contribution in Section 1. Since the proposed benchmark datasets only drive from existing ones with some minor adaptions (translation?).
- For decoder-only models, in addition to BLOOM, I also recommend the authors conduct experiments on BLOOMZ and LLaMA. (1) For BLOOMZ, it can further analyze the influence of fine-tuning with human instruction (STF) on cross-lingual consistency. (2) For LLaMA, it could be better to conduct experiments on the open-source cutting-edge LLMs and convince the conclusions.

**Reproducibility:**

4: Could mostly reproduce the results, but there may be some variation because of sample variance or minor variations in their interpretation of the protocol or method.

**Reviewer Confidence:**

4: Quite sure. I tried to check the important points carefully. It's unlikely, though conceivable, that I missed something that should affect my ratings.

---

> ### Author Rebuttal · Authors · 2023-08-27
>
> Thank you very much for the helpful comments!
>
> 1)We agree about the overstatement of the dataset contribution. This was an oversight on our part and will be removed from Section 1. We will make it clear that BMLAMA is a balanced version of existing factual knowledge benchmarks, as we already did in the rest of the paper.
>
> 2)We appreciate your suggestions about involving more decoder-only models. We have managed to evaluate BLOOMZ-3b and LLaMA-7b on the BMLAMA-17 benchmark.
>
> 2.1) For BLOOMZ-3b: The probing accuracy (23.87%) is lower than BLOOM-3b (25.97%), suggesting human instruction fine-tuning has a mildly negative effect on factual knowledge. Nevertheless, average cross-lingual consistency (CLC) remains 33%, and the heatmaps are nearly the same as BLOOM-3b, since they use the exact same tokenizer. This further strengthens our conclusion that CLC is a property independent of accuracy and mostly relies on the subword overlapping.
>
> 2.2) For LLaMA-7b: To ensure a fair comparison between LLaMA and BLOOM, we have also carried out experiments on BLOOM-7.1b, which contains a similar number of parameters. In terms of average factual knowledge accuracy, LLaMA-7b (28.51%) only slightly outperforms BLOOM-7.1b (27.82%). For average CLC, LLaMA-7b reaches 36% and BLOOM-7.1b reaches 34%.
> In addition, we calculate the Pearson correlation scores between CLC and vocabulary overlapping of these two models.
>
> ————————————————————————
>
> | $\quad$ $\quad$ $\quad$ $\quad$ $\quad$$\quad$| Voc (BMLAMA) | $ $ Voc (Flores) $ $ |
>
> | $\quad$LLaMA-7b$\quad$$\quad$| $ $ $ $ $ $ 0.60 (2e-14) $ $ $ $ | $ $ 0.52 (1e-10) $ $|
>
> | $\quad$BLOOM-7.1b$\quad$| $ $ $ $ $ $ 0.69 (4e-20) $ $ $ $ | $ $ 0.48 (3e-09) $ $|
>
> ————————————————————————
>
> Thus, the results on LLaMA also support our finding that CLC has a positive correlation with the overlapping of sub-word vocabulary.
>
> We will add these new results to Table 2 in the paper.

---

### Official Review · Reviewer_Ukqz · 2023-08-07

**Soundness:** 5

**Excitement:**

4: Strong: This paper deepens the understanding of some phenomenon or lowers the barriers to an existing research direction.

**Paper Topic And Main Contributions:**

This paper proposes a cross-lingual factual consistency metric that goes beyond accuracy. Using that metric, the author provide an in-depth analysis that looks at different factors, and spans multiple models and languages. They shed light on how factual knowledge propagate across languages and expose new implications that could be directly leveraged to improve the transferability of MLMs.

**Questions For The Authors:**

-  Line 227-228: Translate what to what? how to ensure the translations are correct and unbiased across languages?

**Reasons To Accept:**

- This is the first paper that conducts an in-depth analysis of factual consistency across languages and makes good use of it to understand how knowledge gets propagated or not across languages and the underlying factors behind that.
- Comprehensive analysis that spans multiple state-of-the-art PLMs including encoder-only, decoder-only, encoder-decoder and a representative set of typological languages to draw conclusions.
- I think the way they designed this evaluation for factual consistency should be taken as standardized for any paper working on a similar problem.

**Reasons To Reject:**

I couldn't come up with any reasons to reject it! This is solid work in my opinion :)

**Reproducibility:**

4: Could mostly reproduce the results, but there may be some variation because of sample variance or minor variations in their interpretation of the protocol or method.

**Reviewer Confidence:**

3: Pretty sure, but there's a chance I missed something. Although I have a good feel for this area in general, I did not carefully check the paper's details, e.g., the math, experimental design, or novelty.

**Typos Grammar Style And Presentation Improvements:**

- make Figures 3 and 4 more readable. The font for the numbers is too small.

---

> ### Author Rebuttal · Authors · 2023-08-27
>
> Thanks for your time and effort in reviewing our paper!
>
> 1)Regarding your question about the translation, the probing queries and candidates for each query were originally written in English and then translated to other non-English languages for probing multilingual factual knowledge.
>
> As for your concern about the correctness and bias issue, our BMLAMA is derived from X-FACTR (Jiang et al., 2020) and mLAMA (Kassner et al., 2021), where all translated queries were manually checked and corrected to ensure each query contains the slots for the subject and object exactly once. Moreover, according to Kassner et al. (2021), manual calibration of the machine translated queries (e.g. fixing grammatical errors or revising by native speakers) had little effect on the probing performance.

---

### Meta-Review · Area_Chair_DMD5 · 2023-09-19

**Recommendation:** 5

**Metareview:**

The reviewers are in agreement that this is an excellent submission. It is the first study of factual consistency across languages, and provides valuable analyses that are deep and comprehensive.

---

### Decision · Program_Chairs · 2023-10-07

**Decision:**

Accept-Main

**Comment:**

The reviewers are in agreement that this is an excellent submission. It is the first study of factual consistency across languages, and provides valuable analyses that are deep and comprehensive.